# Sequence Similarity Network Analysis Provides Insight into the Temporal and Geographical Distribution of Mutations in SARS-CoV-2 Spike Protein

**DOI:** 10.3390/v14081672

**Published:** 2022-07-29

**Authors:** Shruti S. Patil, Helen N. Catanese, Kelly A. Brayton, Eric T. Lofgren, Assefaw H. Gebremedhin

**Affiliations:** 1School of Electrical Engineering and Computer Science, Washington State University, Pullman, WA 99164, USA; shrutisunil.patil@wsu.edu (S.S.P.); helen.catanese@wsu.edu (H.N.C.); kbrayton@wsu.edu (K.A.B.); 2Department of Veterinary Microbiology and Pathology, Washington State University, Pullman, WA 99164, USA; 3Paul G. Allen School for Global Health, Washington State University, Pullman, WA 99164, USA; eric.lofgren@wsu.edu

**Keywords:** sequence similarity network, SARS-CoV-2, spike protein, mutations

## Abstract

Severe acute respiratory syndrome-related coronavirus (SARS-CoV-2), which still infects hundreds of thousands of people globally each day despite various countermeasures, has been mutating rapidly. Mutations in the spike (S) protein seem to play a vital role in viral stability, transmission, and adaptability. Therefore, to control the spread of the virus, it is important to gain insight into the evolution and transmission of the S protein. This study deals with the temporal and geographical distribution of mutant S proteins from sequences gathered across the US over a period of 19 months in 2020 and 2021. The S protein sequences are studied using two approaches: (i) multiple sequence alignment is used to identify prominent mutations and highly mutable regions and (ii) sequence similarity networks are subsequently employed to gain further insight and study mutation profiles of concerning variants across the defined time periods and states. Additionally, we tracked the variants using visualizations on geographical maps. The visualizations produced using the Directed Weighted All Nearest Neighbors (DiWANN) networks and maps provided insights into the transmission of the virus that reflect well the statistics reported for the time periods studied. We found that the networks created using DiWANN are superior to commonly used approximate distance networks created using BLAST bitscores. The study offers a richer computational approach to analyze the transmission profile of the prominent S protein mutations in SARS-CoV-2 and can be extended to other proteins and viruses.

## 1. Introduction

The COVID-19 pandemic has caused over 230 million infections globally with a fatality rate of approximately 2% as of September 2021 [1]. The causative virus species was identified as severe acute respiratory syndrome-related coronavirus and was named SARS-CoV-2 [2]. The SARS-CoV-2 spike (S) protein is a homo-trimeric protein which is key to viral entry and consists of two subunits S1 and S2 [3,4,5]. Subunit S1 forms a budding head and attaches to host cell angiotensin-converting enzyme-2 (ACE-2) receptors [6,7]. Subunit S2 has a stalk-like structure that fuses the viral and host membrane after which the viral RNA is released into the host cell. Therefore, the S protein is vital in determining the infectivity and transmissibility of the virus.

Various studies confirm the major role of the S protein in SARS-CoV-2 pathogenesis [5,8]. Hence, mutations in the S protein could have a significant effect on protein stability, viral transmission, adaptability, and diversification [9]. Multiple mutations, deletions, and recombinations have been observed in the S protein as the virus encounters diverse host immune systems, despite the countermeasures across the world [10]. Additionally, the mutation rate for RNA viruses is extremely high, increasing the probability of mutations occurring in the protein. RNA viruses evolve on observable timescales and thus mutations in the virus can be studied over time. The timely study of viral evolution creates opportunities to combat viral diseases [11]. Therefore, the study of evolution of SARS-CoV-2 would be one of the effective ways to provide insights into the regions of the virus to be targeted by vaccines or treatments.

The role of S protein as an antigen and the rising number of mutations in the S protein have resulted in vaccines that focus on targeting this protein [12,13]. Some vaccines focus on the full-length S protein while others use only parts of the protein, mostly the highly immunogenic regions such as the receptor binding domain (RBD) [14,15]. For instance, the Pfizer-BioNTech vaccine is a mRNA vaccine packed as a lipid nanoparticle and works against the S protein. It is of paramount importance to closely monitor evolution of the S protein in the circulating virus. Many mutations have been reported in the S protein but only a few of them persist over time and influence the virus infectivity. The D614G mutation has been reported to be common and is known to increase viral infectivity and transmissibility [16]. Several other variants have also been reported, among which the Delta and Omicron variants are known for their most severe effects [17,18,19].

Variants are commonly studied using phylogenetic tree analysis. The Pango network is a highly regarded tool to identify and name lineages of SARS-CoV-2 and uses a hierarchical system to study the virus evolution [20]. Another popular tool, NextStrain [21], also uses phylogenetic tree analysis to study pathogen evolution [22]. Phylogenetic tree analysis provides a high-level overview of the evolution and spread of the virus. Sequence alignment and phylogenetic trees are good tools to study differences among sequences, however, Sequence Similarity Networks (SSNs) can visualize sequence relationships better and additionally identify complex relationships. These networks enable visualization of large sequence sets and allow for better analysis using customizations such as defining a similarity cut off for sequences [23]. Protein sequences and protein families have been widely studied using SSNs to assist in identifying function of uncharacterized proteins, unexplored families and intermediate steps in evolution [24,25,26,27]. Additionally, SSNs have also been used to study SARS-CoV-2, mainly to determine sequence similarities between SARS-CoV-2 and other viruses [28,29,30]. This work shows how SSNs prove to be good tools to study distribution of variants of the virus.

To understand the distribution of mutations in the S protein, we carried out an extensive analysis of S protein sequences reported in the US in 2020 and 2021. In this work, using the large amount of data available in the NCBI Virus repository [31], we perform multiple sequence alignment and network analysis to identify prominent S protein mutations and study their distribution across the US over time. The analysis examines more than 100,000 sequences collected across the US from January 2020 to July 2021. Our network analysis approach uses a variant of a SSN, where the distance (dissimilarity) between sequences is used to construct the networks [23]. In particular, we use the Directed Weighted All Nearest Neighbor (DiWANN) model [32] to analyze the temporal and geographical distribution patterns of the mutant S proteins.

The DiWANN model was applied on short sequence repeats in a previous study and has proven to be more effective than threshold based networks for short sequences. This study deals with much longer sequences (about 20–25 times longer) comprising of amino acids. Additionally, we apply various network visualization techniques to elucidate transmission of the virus. The results were supplemented by maps created using Tableau to show the distribution of variants across the states. Taken together, the analysis demonstrates a new approach to study the transmission of variants across a geographical area over time.

### 1.1. Spike Protein Structure

The SARS-CoV-2 S protein is 1273 amino acids (aa) in length. It consists of a signal peptide (aa 1-13), the S1 subunit (aa 14-685), and the S2 subunit (aa 686-1273) [5]. The S protein is cleaved into the S1 and S2 subunits at a furin cleavage site during viral infection. This activates the membrane fusion domain allowing the virus to enter the target cells. Subunit S1 comprises an N-terminal domain (NTD: aa 15-261) and a receptor-binding domain (RBD: aa 319-541) that binds to the cell receptor ACE2 [33,34]. The RBD is a critical target for neutralizing antibodies and contains a receptor-binding motif (RBM) which makes direct contact with ACE2 [35,36]. The S2 subunit is comprised of a fusion peptide (FP: aa 788-806), heptapeptide repeat sequence 1 (HR1: aa 912-984), heptapeptide repeat sequence 2 (HR2: aa 1163-1213), transmembrane domain (TM: aa 1213-1237) and cytoplasmic tail domain (CT: aa 1237-1273). S2 is responsible for viral entry and fusion.

### 1.2. Network Analysis of Spike Protein

Mutations in the S protein can be observed by aligning sequences or calculating distance (dissimilarity) between the reported sequences. There are various distance metrics and algorithms to determine similarity, including Levenshtein distance (edit distance) [37,38], Hamming distance [39,40], Needleman-Wunsh [41,42] and Smith-Waterman [43]. These metrics tend to be computationally expensive to calculate. Similarity scores can be obtained from faster heuristic (approximate distance) methods such as Basic Local Alignment Search Tool (BLAST) [44] and its variants [45]. In this work, (exact) edit distance is used to compute the pairwise distance between the S protein sequences, thanks to an efficient implementation available in the DiWANN model we used [32]. We compared the DiWANN networks created to the threshold-based networks created using BLAST scores.

Distance measures are used by SSN-based approaches to compute the dissimilarity between sequences. The most similar sequences are then connected by edges. Sequence similarity can, for instance, be used to identify homologous proteins or genes [46] that share a common evolutionary ancestor and sequences with similar functions [47,48]. In this study, sequence similarity is used to form a network in which nodes represent the S protein sequence and edges indicate dissimilarity (distance) in aa sequence. The network is modeled using the Directed Weighted All Nearest Neighbor (DiWANN) network [32]. In this network representation each sequence is denoted by a node. A directed edge is drawn from every node to the node it is the closest to in terms of edit distance. In scenarios where multiple sequences tie for being the closest to a sequence considered, all the edges are kept, ensuring important structural data is not lost.

The DiWANN network is much sparser than a basic SSN, while still capturing the core structural elements of the sequence dataset [32]. The underlying algorithm to construct DiWANN is relatively simple and uses a pruning and bounding technique for efficient distance calculations. The distance calculations that are not needed are pruned and the calculations that are needed are bounded.

DiWANN has several modeling and algorithmic strengths that make it a preferable approach for constructing SSNs compared to k-nearest neighbors (kNN) and all nearest neighbors (ANN) approaches. The kNN approach relies on selection of a threshold which can be difficult to determine. A threshold too large creates a dense network, which can be computationally expensive to work with, while a threshold too low can lose important structure. The ANN approach avoids these threshold issues, however, computing distances for large sequences can still be prohibitively costly, leading to similarity networks to be built with approximate distance metrics, which can be unreliable. The DiWANN network is both inherently sparse (for most datasets) and does not rely on approximate distance metrics. In this study, we create DiWANN networks to visualize the distribution of prominent mutations across the US in a manner that facilitates analysis of the spread of the virus over time.

## 2. Materials and Methods

### 2.1. Dataset Preparation

In this study, we extracted over 100,000 complete nucleotide sequences for the S protein from the NCBI Virus repository [31]. The SARS-CoV-2 Data Hub provided by NCBI Virus made filtering and downloading desired data convenient. We extracted only the S protein sequences from the repository using the Proteins filter. Additional filters including Geographic Region and Collection Data were useful for data collection. The sequences were reported from US states over a period of 19 months (1 January 2020 to 31 July 2021). A significant portion of the data had repeated sequences, so we extracted only the unique sequences to make the SSNs. A few of the sequences were unique due to ambiguous sequence resulting in unresolved residues (X, any aa).

To better visualize and understand the variations in the protein and the transmission pattern, we divided the dataset into eleven time periods. The divisions were made to correspond to the spread of the virus and measures taken to control the spread in the country. The first time period ranged from 1 January 2020 to 20 March 2020 (T1) and represented the initial stages of the pandemic. The second time period ranged from 21 March 2020 to 30 April 2020 (T2) and represents the stay-at-home order which lasted for more than a month. Many states started reopening in the first week of May. Therefore, we considered the third time period from 1 May 2020 to 20 September 2020 (T3). The fourth time period extended from 21 September 2020 to 31 December 2020 (T4). The fifth time period lasted from 1 January 2021 to 31 January 2021 (T5). There were many new mutations arising every month after T5 and hence, every month has been considered as a time period after T5 for better analysis of distribution of the mutations. Therefore, the month of February 2021 was considered as T6, March 2021 as T7, April 2021 as T8, May 2021 as T9, June 2021 as T10, and July 2021 as T11.

This temporal division allowed for more useful analysis of the transmission profile of the virus in the US compared to an analysis that does not use temporal divisions. Relevant statistics of the sequences in our dataset are provided in Table 1. Along with the unique sequences, we also considered the number of repetitions of each sequence in the SSNs.

### 2.2. Alignment of Spike Protein

We performed multiple sequence alignment (MSA) to observe mutations in the S protein sequences of SARS-CoV2. The alignment tool Clustal Omega [49] was used for MSA, and the tool MView [50] was used to visualize the MSA results. The tool MView reformats multiple alignments by adding HTML markups for coloring and web page layout. It also provides annotation columns such as percent identity, enabling us to identify mutations more conveniently. The mutations were checked within each period as well as across the time periods. Most mutations were found only in a single isolate, that is, they occurred only in one of the analyzed unique sequences. We focused on the mutations that were prevalent over time, or across geographic regions. Some of these mutations were present from T1, but most of them were encountered in the later time periods.

### 2.3. Network Analysis

Sequence alignment tools are ill-suited for visualizing distance (dissimilarity) among all the sequences in the dataset while also taking the temporal and geographical information into consideration. In contrast, sequence similarity networks can provide meaningful insights into the relationship between the sequences and can also be used to depict temporal and geographical distribution. SSNs are networks in which the nodes are sequences and edges depict a closely related pair of sequences. This class of networks can aid biologists in finding useful starting points for analyzing sequences belonging to different groups or families [23]. In this work, we use a variant of SSN, the DiWANN network. To analyze and visualize the networks we used the igraph package in Python [51].

We created a DiWANN network for the sequences in each time period. The nodes are colored to indicate the geographical source of the sequences. In particular, we color the sequences reported from the same geographical area (state) the same color. Since some states had only a few samples reported, we colored only the states that had sufficient data. Consequently, the colored states are Florida, Washington, California, Wisconsin, Massachusetts, Michigan, and New York. The other states are simply divided into East and West. We consider the states to the west of Minnesota as the West region and the rest as East. There were a few sequences without a state label; we included these in our analysis as they still contain pertinent information about the sequences in the country. Those sequences have been labeled as ‘USA’ for the purpose of coloring the nodes.

An important note is that the location assigned to a node (sequence) is the state where the sequence first occurred. There could be more than one state where a sequence was reported, but for this network analysis, we considered only the first occurrence. However, the count of the occurrence of each sequence is taken into consideration, utilizing the whole dataset. The entire dataset is also considered while visualizing the distribution of variants across the US.

Both the temporal and geographical distribution of the prominent mutations obtained from the sequence alignment were studied using the DiWANN networks. A network was created for each time period. The individual time period networks were visualized to show the location and the variant using different node colors and node shapes, respectively. We supplement the SSN models with geographic maps for each time period, showing where particular spike protein mutations appeared across the country over time. The maps were created using Tableau [52].

Finally, we create SSNs using a threshold-based approach and compare them to the DiWANN networks. The thresholds for creating the SSNs are the bitscore values obtained by using Basic Local Alignment Search Tool (BLAST) [53]. The BLAST tool is a program to find regions of similarity between biological sequences. There are many variations of BLAST, and for our study we have made use of Protein BLAST (blastp).

## 3. Results

### 3.1. Variations in Spike Protein

We encountered more than 400 mutations while performing MSA of the S protein sequences. The mutations in T1 were mostly found in single isolates. T2, T3, T4 and T5 had a significant number of mutations that occurred in more than one sequence, and periods T6 and later had many occurrences of several mutations. The earliest most notable variant in the S protein was D614G (aspartic acid at residue 614 replaced by glycine). Residue 614 of the spike protein is part of the carboxy(C)-terminal region of the S1 subunit and belongs to the region that directly associates with S2. This variant occupied a large portion of detected mutants in the US since March 2020.

Other potentially significant mutations observed in the first three time periods include L5F, V6F, S221L, A570V, and P1263L as shown in Figure 1a. Most of these mutations occurred only in a single isolate in T1, but the number of sequences containing these mutations increased in T2 and T3. Although these five mutations were detected in T4 and later time periods, most were not prevalent. Only mutation L5F persisted in all time periods with high occurrences after T4. The other mutants occurred only a few times in the later time periods, with S221L and P1263L having higher occurrences in T2 and T3.

We observed several new mutations in T4 and T5 sequences including S13I, W152C, V382L, L452R, Q677H and P681H as shown in Figure 1b. The number of mutations as well as the frequency of occurrence of the mutations increased in period T6 and later. The most common mutations in T6, T7 and T8 are shown in Figure 1c and common mutations in T9, T10 and T11 can be seen in Figure 1d. Most of the mutations occurred in more than 50 unique sequences, which seems to be a significant number of sequences considering that most of the mutations occurred in a single isolate. Additionally, many mutations co-occur with each other. Most of the mutations shown are characteristic mutations of the four common SARS-CoV-2 variants: Alpha, Beta, Delta and Gamma. There are a few mutations that occur in more than one variant as shown in Table 2.

We found over 30 mutations in the receptor binding domain (RBD). Among these, A344S, V483A and A522V were dominant in the first 3 time periods T1-T3, but were soon dominated by the mutations V382L, L452R and A520S in T4 and T5. Mutation L452R became more common after T5 and got characterized as an important mutation in the Delta variant. Additionally, new mutations became dominant after T5, including K417T, S477N and S494P in T6, T7 and T8, and R346K, T478K, E484K and N501Y in T9, T10 and T11. Period T11 had a significant increase in several mutations, including K77T, G142D, A222V, T478K, T859N and D950N, most of which are characteristic mutations of the Delta variant. Overall, the number of mutations, especially in subunit S1 increased after T5.

### 3.2. Distribution of Highly Transmissible Variants

The results from MSA show that the D614G variant became well established by the period T3 and the Delta variant spread rapidly across the US. The DiWANN networks we constructed for the D614G and Delta variants provide insight into the transmission trend of the variants over time and across states. The networks also provide relative occurrences of sequences depicted via node sizes. Therefore, the central nodes in the large components (cores) of the networks are larger in size compared to the surrounding nodes due to high occurrences. These nodes usually represent the D614G or Delta variant sequences. The D614G variant spread very quickly, and glycine (G) has become the most prevalent aa at residue 614 over time as shown in the networks in Figure 2.

At the start of the pandemic, in T1, there was a similar distribution of the two variants, but G started increasing in representation in T2, and became the dominant variant by T3. The networks corresponding to T1 and T2 had two cores, which represented the two variants (D and G at residue 614), however, the networks for T3 and T4 have just one large component representing aa G at residue 614. The emergence of the variant can be observed well in these 4 time periods, after which there is almost no occurrences of the D variant. Since we colored the nodes depending on the state in which the sequence first occurred, the networks depict the transmission pattern in the country over time.

While performing MSA, we observed that the number of mutations and the frequency of these mutations started to rise in 2021. The four variants: Alpha, Beta, Delta and Gamma occurred in significant numbers in period T6 and later. Figure 3 shows the occurrence frequency of the variants. As seen in the figure, the frequency of the variants has been rising since T6, with the maximum count in T9 according to the reported data. The figure also indicates that the Delta variant has been the most common since T9 (May 2021), whereas the occurrence of other variants has been decreasing since T9.

The bar plots in Figure 3 show us the occurrence pattern of the variants when we consider only the unique sequences. The bar plots also indicate the number of mutant versions. There are many mutations in the variants, in addition to the characteristic mutations. To see a more detailed distribution of the variants across the states, we consider the whole dataset, and not just the unique sequences. This ensures the analysis of all the occurrences of the variants and not just the first occurrence. The geographical maps showing the distribution of the four variants in the periods T6, T8, T9 and T11 are provided in Figure 4. Similar maps for the periods T7 and T10 are provided in the Appendix A.

The Centers for Disease Control and prevention (CDC) had classified variants Alpha, Beta, Delta and Gamma as variants of concern (VOC). However, CDC downgraded Alpha, Beta and Gamma to variants being monitored (VBM) in September 2021; Delta was downgraded to a VBM in April 2022. The distribution of the Delta variant in the US during periods T6–T8 can be seen in the DiWANN networks in Figure 5; analogous information for the periods T9–T11 is given in Figure 6.

The DiWANN networks consider only the unique sequences, that is the first occurrence of the sequences. This provides us with information about the sequences and the mutations in the spike protein. However, to analyze the distribution of the Delta variant in the US, we have created geographical maps for the time periods T6, T8, T9 and T11 that utilize the entire dataset and not just the unique sequences. Figure 7 shows us the distribution of the Delta variant in T6, T8, T9 and T11, and Appendix A shows us the distribution of the Delta variant in T7 and T10.

As motivated in the Introduction, we use the network model DiWANN because it is better suited for our dataset and analysis compared to alternative SSN models. To verify this, we compare it against the use of SSN based on threshold values and scores using BLAST [53]. We created threshold-based networks (TBNs) for three time periods, namely, T6, T8 and T11 as they seemed good breakpoints in the study of the temporal distribution of the mutations in the S protein. Bit score was used as a threshold value to create the SSN and the best threshold was obtained by trial. A threshold that did not produce many singleton nodes and at the same time did not make the network too dense was selected. The bitscores selected for the networks representing T6, T8 and T11 were 2633, 2632 and 2628, respectively. The networks created are shown in Figure 8 and the network properties are summarized in Table 3.

## 4. Discussion

In this study, we analyzed SARS-CoV-2 S protein mutations and studied the temporal and geographical distribution of some highly transmissible variants in the US. The logical division of the dataset into time periods depicted a story of how the S protein in the virus evolved over time, despite the counter measures taken around the world. We initially divided the time periods by month, which was a finer division for the year 2020, however, we found that dividing the year according to the stages in the pandemic gave us better insights into viral transmission across the country. Additionally, during the early months of the pandemic, there were not many variants and so the chosen time periods depicted a better story.

The MSA results indicate that the virus mutated faster in the later stages of the pandemic than in the earlier stages, making it of paramount importance that we keep track of the changes that are potentially increasing the viral infectivity and transmissibility. Further, analysis using MSA showed that there have been more mutations since T6 (February 2021), which corresponds to the deployment of vaccines, indicating that the virus is adapting to selective pressure. Additionally, more heavily mutated SARS-CoV-2 variants have been emerging over time [54]. Overall, the S1 subunit has more mutations than the S2 subunit, possibly because it attaches to the host through the RBD. There are mutations in the RBD that have occurred persistently in large numbers of sequences. Most notable ones include L452R, E484K, and N501Y, and these have been found to have influence on the virus’ fitness [55]. Considering such persistent mutations while developing vaccines or treatments can be beneficial.

Variant D614G had been established by the end of 2020 as can be seen from Figure 2. The figure also depicts how the variant evolved over time in the US. The node colors representing the states in the T1 network clearly show the dominance of triangular nodes, which are sequences with aa D at residue 614, in the western states. The other component has very few nodes from Washington and is primarily comprised of circular nodes which are sequences with G at residue 614 from Florida, New York, and other eastern states. The figure shows how the 614D version of the virus came into western US while the 614G version entered on the eastern coast before becoming the dominant variant nationwide.

The networks for T3 and T4 have a single large component primarily represented by the variant D614G across the states. Studies suggest an increase in viral infection as a result of the mutation at residue 614 due to the reduction in shedding of S1 and an increase in the incorporation of the total S protein into the virus [16]. This could have led to higher transmissibility and infectivity of the D614G variant. By the end of 2020, due to rapid transmission and dominance, the G614 variant had become the reference to study other mutations.

Other important variants were studied in a similar manner. From Figure 3, we see that, out of the four common variants, Delta persists to be the most infective and transmissible by the end of July 2021. The high number of distinct sequences of the Delta variant in the bar plot also indicates that the variant has been mutating significantly in addition to the characteristic mutations. However, most of the additional mutations do not seem concerning as they occur in relatively low number of sequences, sometimes just in a single isolate.

The temporal distribution of the Delta variant shows how the occurrence of this variant increased over time and has been high since T9 (May 2021). The T9, T10, and T11 networks in Figure 6 seem to be more clustered than the T6, T7, and T8 networks in Figure 5, suggesting the high occurrence of the Delta variant. The Delta variant was reported to represent only 0.1 percent of the cases in the US in April 2021, but the variant accounted for 1.3 percent of cases in May, and by early June, that number jumped to 9.5 percent [56].

Another interesting observation about the distribution of Delta variant is that the variant was initially found more in the west (notably in California) and became dominant across the US by T11. The T6 network in Figure 5a shows that most of the inverted triangular nodes that represent the Delta variant are from California while the other two networks representing T7 and T8 in Figure 5b,c, respectively, have a mixed distribution of sequences from both east and west states. The networks in Figure 6 show an increase in Delta sequences, especially in the east states. A notable observation in Figure 6c showing the T11 network is the component having the Delta variant nodes mainly from the east states. This indicates that these sequences are more similar to each other than the other sequences which could be because of similar virus reaction to the regional vaccination or weather conditions.

The geographical maps in Figure 7 show a similar distribution, with a higher portion of the Delta variant in the west in T6, especially in California. In the subsequent periods, we observed the number of Delta sequences decreased in California and increased in the south east states such as Florida and Louisiana. The observed trends align with the news reported at the end of summer 2021 [57,58]. This demonstrates that the approach used in this study is consistent with the way the virus transmitted across the country. A summary of some of the key observed trends for all the periods is provided in Figure 9.

Our study involved a large number of S protein sequences collected over a period of 19 months. The data provided significant insights into the transmission and mutations of the virus. The approach used is useful in deriving information to be considered while targeting regions of the S protein for treatments and analyzing pathogenesis of the virus. However, a limitation of this analysis is that the completeness of the data obtained from NCBI Virus depends on the states reporting the sequences. There were a good number of sequences from most highly populated states, but some states reported only a few sequences or did not report consistently with time. Therefore, our results rely on the available data, but we believe they provide a picture of the temporal and geographical distribution of mutations in the spike protein across the US.

The study demonstrates how an approach using the DiWANN model and analysis can be effective in providing insights about the transmission of a virus. We applied this approach to identify trends over time periods (January 2020–July 2021) that had some established facts regarding the virus, enabling us to verify that our approach provided valid results. Additionally, studying the pandemic right from the start till July 2021 provided us with a broad picture of the spread of different variants, especially the Delta variant, which was the most severe variant among people who were not vaccinated. This effective approach combines temporal and geographical data and can potentially be used to study other variants of SARS-CoV-2 or other infectious diseases in future works.

We chose a DiWANN SSN as our model for this analysis over a threshold-based SSN using either an exact or approximate similarity metrics because it offered better results while keeping computational cost low; in particular, exact threshold-based SSN is overly costly to construct and approximate threshold-based SSN can provide inferior results [32]. Additionally, both methods can suffer from a poorly chosen threshold. In our study we have chosen a threshold by trial such that there is a balance between the density of the network and the number of singleton nodes. The constructed threshold-based networks using BLAST bitscores were usually denser than the corresponding DiWANN networks as can be seen in Figure 8 and Table 3. Moreover, the threshold-based SSNs have singleton nodes.

We found DiWANN to be better suited for this dataset which consists of quite similar sequences as all are spike proteins and fairly conserved. However, although the DiWANN network is much faster to construct than a threshold-based SSN using an exact similarity measure, the approach may still be computationally intensive for large datasets. In such cases, some approximations may become necessary.

## 5. Conclusions

With the continuously growing outbreak of SARS-CoV-2, understanding the biology of the infection is of paramount importance. Study of the spike (S) protein and its mutations is key to understanding viral transmission and pathogenicity. In this study, we combine two approaches to study mutations in the S protein, namely, multiple sequence alignment and network analysis. Prominent mutations were identified by performing MSA on the collected sequences from the NCBI Virus repository. The temporal and geographical distribution of two important variants, namely the D614G variant and the Delta variant were studied using a variant of sequence similarity networks called DiWANN. Each stage of the pandemic has its own story which we observed in the DiWANN networks, verifying the soundness of the implemented approach. Our computational approach provided richer insights into the behavior of the virus over time and can be used to gain insights about other variants, proteins or viruses.

## Figures and Tables

**Figure 1 viruses-14-01672-f001:**
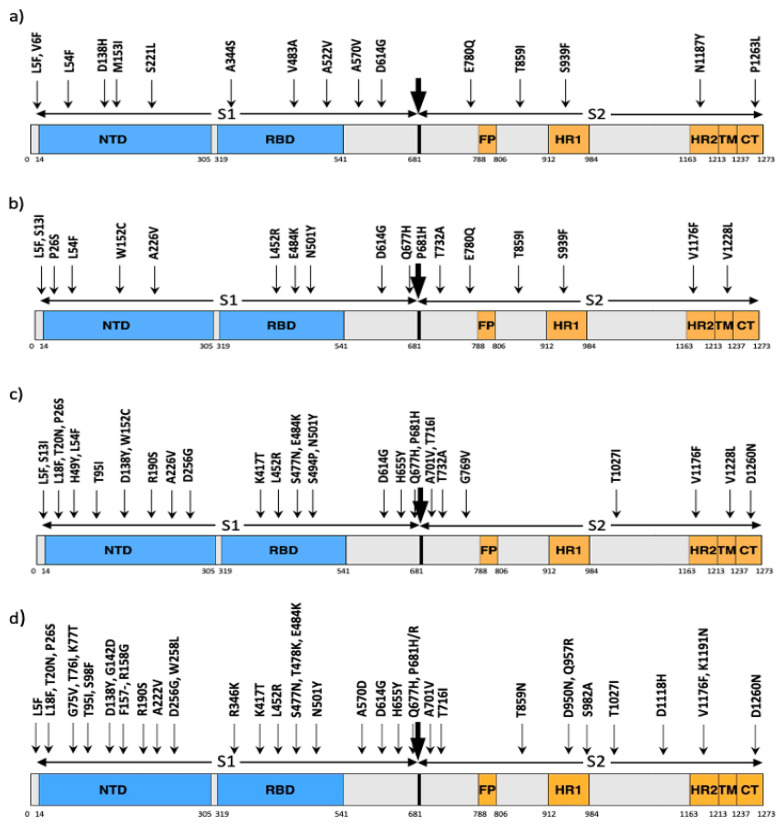
Notable mutations in the S protein of SARS-CoV-2. The arrows show where the mutations took place. (**a**) Mutations that were persistent over T1, T2 and T3. (**b**) Mutations that occurred in a significant number of isolates in T4 and T5. (**c**) Mutations that occurred in a significant number of isolates in T6, T7 and T8. (**d**) Mutations that occurred in a significant number of isolates in T9, T10 and T11.

**Figure 2 viruses-14-01672-f002:**
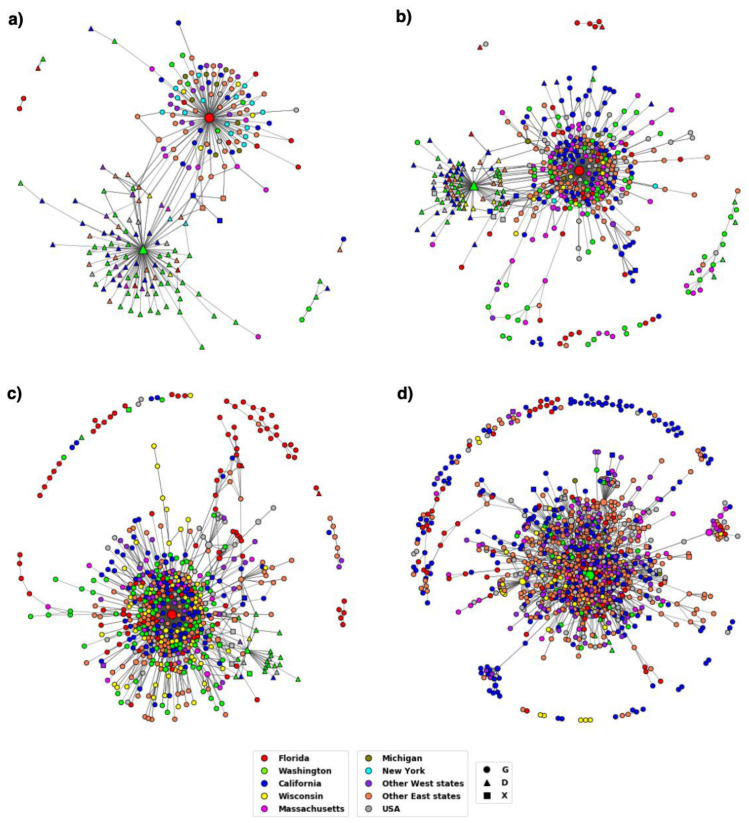
DiWANN networks for the first four time periods T1–T4. (**a**) The network for T1 comprised of 221 nodes, (**b**) T2 comprised of 456 nodes, (**c**) T3 comprised of 622 nodes, (**d**) T4 comprised of 1035 nodes. The shape of the nodes corresponds to the aa at residue 614 and the node color to the location as depicted by the legend. Nodes labelled as ‘USA’ did not have location information.

**Figure 3 viruses-14-01672-f003:**
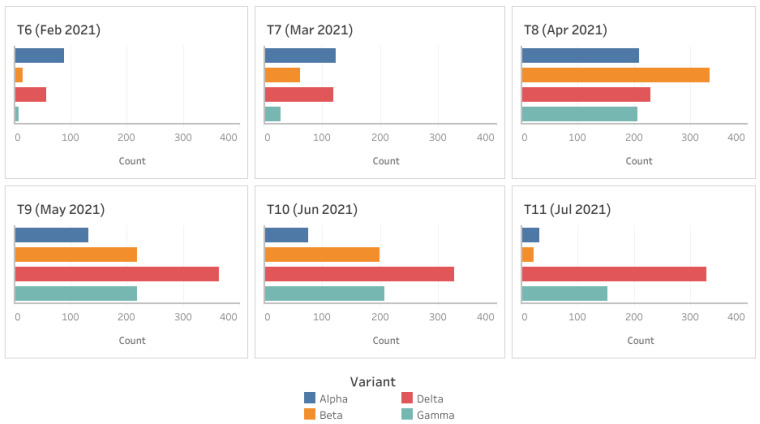
Occurrence of the four variants: Alpha, Beta, Delta and Gamma in periods T6 to T11. The variants are color coded as shown by the legend.

**Figure 4 viruses-14-01672-f004:**
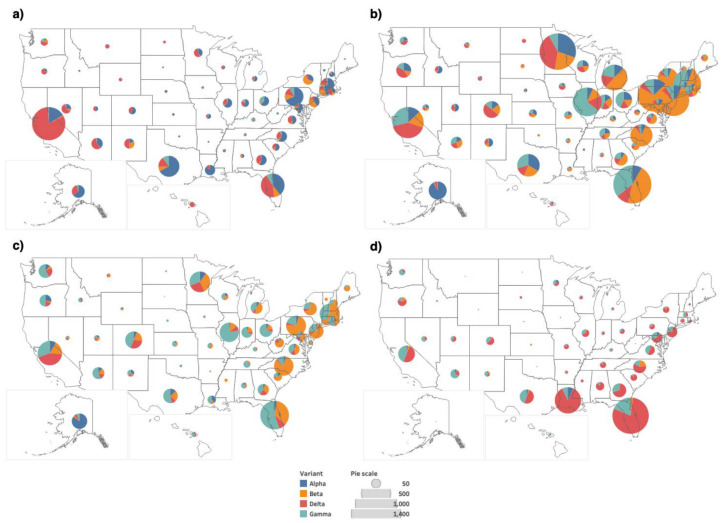
Geographical distribution of the Alpha, Beta, Delta and Gamma variants across the US for four time periods, (**a**) T6, (**b**) T8, (**c**) T9, and (**d**) T11. The legend shows the colors representing the four variants and the pie scale used to size the pies. The pie scale shows count of occurrences, and the pie is sized correspondingly.

**Figure 5 viruses-14-01672-f005:**
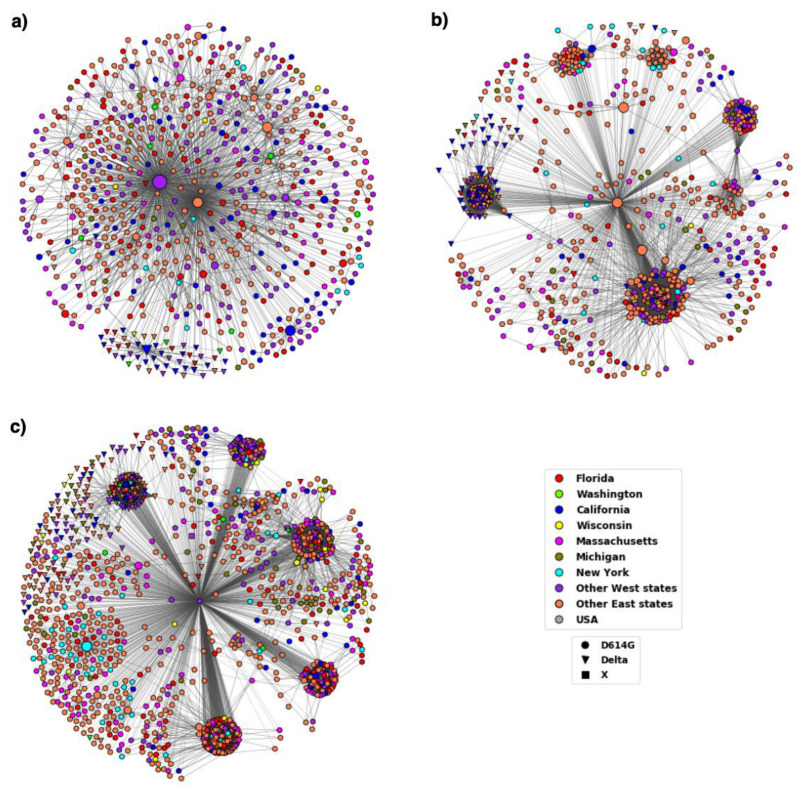
DiWANN networks for periods T6–T8. (**a**) The network for T6 comprised of 688 nodes, (**b**) T7 comprised of 736 nodes, (**c**) T8 comprised of 1478 nodes. The shape of the node corresponds to the variant and the node color to the location as depicted by the legend. Nodes labelled as ‘USA’ did not have location information. The inverted triangular nodes have characteristic Delta variant mutations whereas the circular nodes might have other mutations in the established D614G variant.

**Figure 6 viruses-14-01672-f006:**
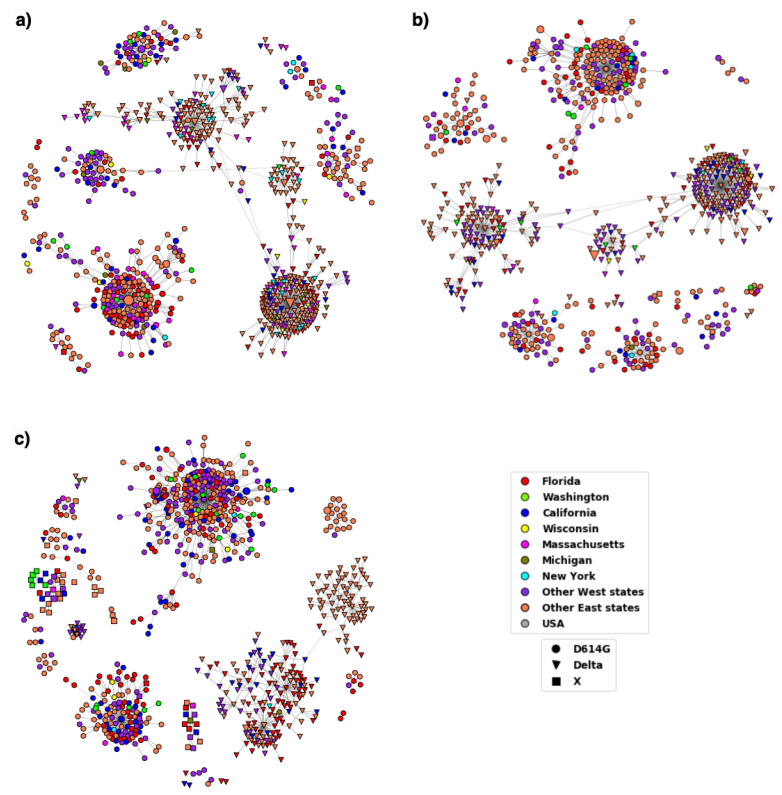
DiWANN networks for the last three time periods T9–T11. (**a**) The network for T9 comprised of 762 nodes, (**b**) T10 comprised of 700 nodes, (**c**) T11 comprised of 786 nodes. The shape of the node corresponds to the variant and the node color to the location as depicted by the legend. Nodes labelled as ‘USA’ did not have location information. The inverted triangular nodes have characteristic Delta variant mutations whereas the circular nodes might have other mutations in the established D614G variant.

**Figure 7 viruses-14-01672-f007:**
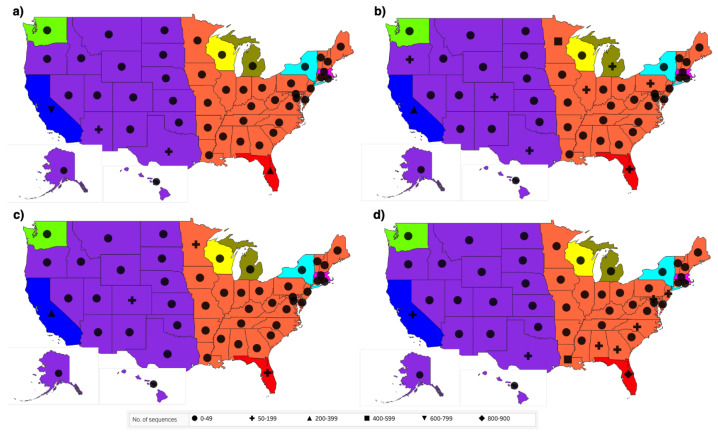
Geographical distribution of the Delta variant across the US for four time periods: (**a**) T6, (**b**) T8, (**c**) T9, and (**d**) T11. The legend shows the shapes representing the count of sequences that have characteristic mutations of the Delta variant. The states have been colored to correspond to the color of the states in the DiWANN networks.

**Figure 8 viruses-14-01672-f008:**
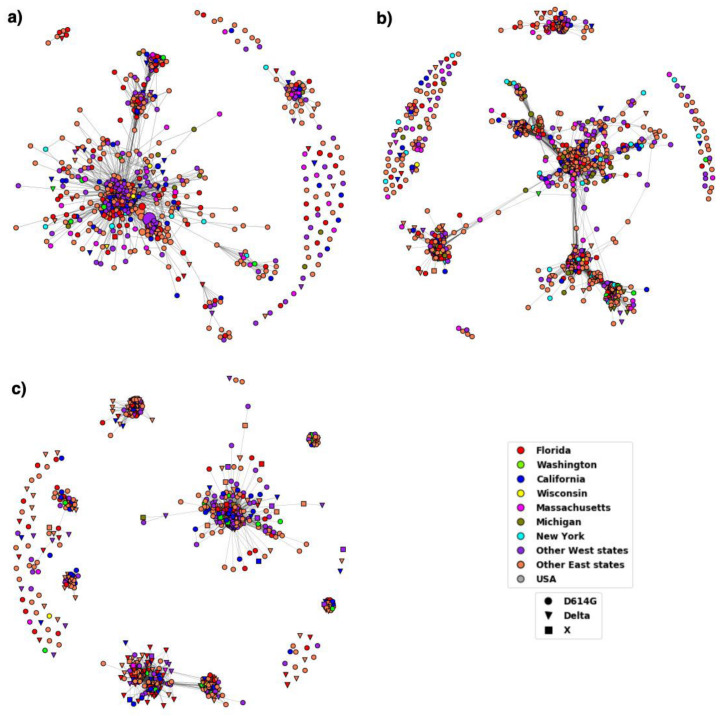
Threshold based networks showing the Delta variant for three time periods. (**a**) The network for T6 comprised of 689 nodes, (**b**) T8 comprised of 1479 nodes, and (**c**) T11 comprised of 786 nodes. The shape of the nodes corresponds the variant and the node color to the location as depicted by the legend. Nodes labelled as ‘USA’ did not have location information. The inverted triangular nodes have characteristic Delta variant mutations whereas the circular nodes might have other mutations in the established D614G variant.

**Figure 9 viruses-14-01672-f009:**
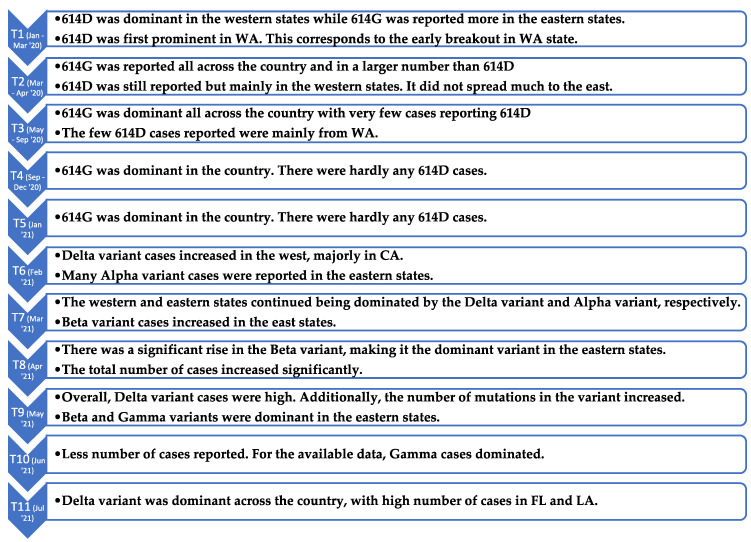
Timeline summarizing the observed trends in the time periods T1–T11.

**Table 1 viruses-14-01672-t001:** Dataset description.

Time Period	Total Number of Sequences	Number of Unique Sequences	Number of States Reported
T1: 1 Jan 2020 to 20 Mar 2020	4047	221	48
T2: 21 Mar 2020 to 30 Apr 2020	5384	456	37
T3: 1 May 2020 to 20 Sep 2020	5876	622	30
T4: 21 Sep 2020 to 31 Dec 2020	5379	1035	49
T5: 1 Jan 2021 to 31 Jan 2021	2932	787	46
T6: 1 Feb 2021 to 28 Feb 2021	17,112	688	49
T7: 1 Mar 2021 to 31 Mar 2021	26,375	736	50
T8: 1 Apr 2021 to 30 Apr 2021	54,883	1478	50
T9: 1 May 2021 to 31 May 2021	31,815	762	49
T10: 1 Jun 2021 to 30 Jun 2021	8598	700	48
T11: 1 Jul 2021 to 31 Jul 2021	5732	786	47

**Table 2 viruses-14-01672-t002:** Variants and their characteristic mutations.

Variant	Mutations
Alpha	∆69, ∆70, ∆144, E484K, S494P, N501Y, A570D, D614G, P681H, T716I, S982A, D1118H, K1191N
Beta	D80A, D215G, ∆241-243, K417N, E484K, N501Y, D614G, A701V
Delta	T19R, V70F, T95I, G142D, ∆156-157, R158G, A222V, W258L, K417N, L452R, T478K, D614G, P681R, D950N
Gamma	L18F, T20N, P26S, D138Y, R190S, K417T, E484K, N501Y, D614G, H655Y, T1027I

**Table 3 viruses-14-01672-t003:** Network properties of DiWANN and threshold-based networks (TBN) for periods T6, T8 and T11.

Network Property	T6		T8		T11	
	DiWANN	TBN	DiWANN	TBN	DiWANN	TBN
Nodes	689	689	1478	1478	786	786
Edges	2408	12,519	67,985	56,006	1995	21,215
Avg. degree	6.98	36.39	91.93	75.78	5.07	53.98
Max degree	688	265	1478	276	353	230
Diameter	9	10	18	11	8	7
Clustering coeff.	0.01	0.64	0.76	0.78	0.1	0.81
No. of Comp.	1	59	1	80	21	67
Largest Comp.	689	569	1478	1217	296	280
Singleton nodes	0	4	0	64	0	56

## Data Availability

All sequence data analyzed here are publicly available at NCBI Virus (https://www.ncbi.nlm.nih.gov/labs/virus/vssi/#/).

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
