# Peer review of "Sequence Similarity Network Analysis Provides Insight into the Temporal and Geographical Distribution of Mutations in SARS-CoV-2 Spike Protein"

_viruses, 2022, doi:10.3390/v14081672_

Round 1
Reviewer 1 Report
The manuscript “Sequence similarity network analysis provides insight into the temporal and geographical distribution of mutations in SARS-CoV-2 spike protein” investigated the temporal and geographical distribution of mutant S proteins using sequences gathered across the US for 19 months. The manuscript is well written and the approach the authors proposed is sound. However, the datasets the authors used is about one year ago and some statements are outdated. Here are my specific comments:
1) The dataset used in the manuscript is from January 1, 2020 to July 31, 2021. That’s about one year ago. It does not reflect the current state of mutations in SARS-CoV-2 spike protein. For example, currently the most dominant strain is Omicron and the paper never discuss that. I would suggest the authors update the data or at least discuss how the dataset one year ago provides insight for the current mutation of omicron variants.
2) In addition to the dataset, many statements in the manuscript are outdated. For example, delta variant is no longer classified as VOC. The authors need to review their manuscript again and keeps the data and these statements updated.
3) The authors did not mention how multiple sequence alignment was used to observe mutations. Could the authors elaborate how the mutation was observed?
Reviewer 2 Report
The paper is very well organized and clearly expresses the concept "big data analysis" is biological relevance.
line 15: delete "the"
line 50: certainly the S portion is relevant in the patient host iteration but at the treatment level other portions especially replicases are also critical it is necessary to correct and explain this concept better.
line 105: add d to appear
line 138: were all sequences downloaded and then used only those from the spike portion? not very clear
lines 147 to 162: does changing the number of periods change the distribution of nodes? by tightening them is it possible to identify "early clusters"?
line 273: merge variants all in capitals.
line 356: there are references on the biological impact and then of vaccines on these mutations is it possible to mention some of them.
line 378: change from is to In
Reviewer 3 Report
> Summarize the study:
The manuscript by Shruti and collaborators describes the use of sequence similarity network (SNN) analysis, in particular the Directed Weighted All Nearest Neighbors (DiWANN), to study the temporal and geographical distribution of mutations in the S protein of SARS-CoV-2 variants. Multiple sequence alignment (MSA) is the default method to study protein variability, and it can be used to identify "hot-spots" for mutations across multiple sequences of the same protein. However, the authors argue that MSA is not suitable for the visualization of complex relationships between sequences. As an alternative, the authors rely on SNN analysis, based on (exact) pairwise distances computed between S protein sequences. Their implementation overcomes limitations related to the computational cost of distance metrics, the imprecision of approximate distances, and the production of overly dense networks by some SNN methods (which complicates the subsequent analyses). The study is relevant and timely, and the manuscript is well written and well organized. However, the main contribution of the manuscript seems unclear. The title, the abstract, the introduction and the conclusion sections hint at "insights" into the SARS-CoV-2 pandemic, but these insights are buried in the results and are not highlighted or discussed in a broader perspective. It sounds like the application to S protein is just a toy example to demonstrate the usefulness of the computational analysis employed, and that the particular biological insights on are not as relevant. That would be fine. But in that case the main contribution would be the computational methods, and the authors fail to clarify what was their specific contribution in that regard. Is the use o SNNs for this type of study innovative or standard in the field? Is this the first time DiWANN was used for this application? Were there open problems in the field that were overcome here for the first time, through the author's unique implementation? Some revision is needed to clarify these points and highlight the main contribution of the manuscript.
> Comments to the authors:
1) If the biological insights on the SARS-CoV-2 pandemic derived from this study are relevant, at least one of them (or a summarized overview of them) should be explicitly described in the Abstract and at the Conclusion section. The authors should make it more clear if the main contribution of this study lies on the biological insights obtained, or in the innovative computational methods, or both.
2) The authors should revise their introduction to make it more clear if the use of SSN for this type of study is something relatively new and uncommon, or if it is already well established in the field. The sentence that introduces the acronym SNN has no references. And the remaining of the introduction offers only one - somewhat old - reference (Atkinson, 2009). A quick search on pubmed for the terms "'sequence similarity networks' AND protein" returns over 50 references more recent than that. None of those are relevant for this study? Broader than that, what is the main contribution of this study in terms of the computational methods? What is standard use of SNNs for protein analysis and what is new ground explored by this study? In particular, the use of DiWANN model for this type of analysis was described by the authors in a previous publication (Catanese et. al., 2018). Is the present manuscript a new application example of the methods previously described, or is it a step forward in refining the use of DiWANN model for this type of study?
3) The second paragraph of section 1.2 has two sentences (lines 112 to 117) explaining the same point, that the nodes are sequences and the edges indicate similarity. I believe the difference is that in the DiWANN case the edges are edit distance. The authors should revise these sentences to clarify the distinction they want to make, and reduce the impression of repeated information.
4) Check for consistent use of acronyms (e.g., KNN and k-NN).
5) In Figure 2, and subsequent images of DiWANN networks, what is the meaning of unconnected nodes?
6) In Figure 2, and subsequent images of DiWANN networks, it is hard to make sense of the color patterns. This color scheme is also reflected on the maps on Figure 7. Why were 6 states represented with different colors, and the rest aggregated into two big blocks? There is also no clear pattern in the colors. Like both New York and California are shades of blue, although no relationship is implied by this choice of colors. If possible, it would facilitate interpretability if the colors reflected a gradient from West to East. It is OK if the 6 states have unique colors while the others belong to two big blocks, but the colors of California and Oregon should be closer to the block of West states. While the states around the great lakes should have colors more similar to each other and to the East states. Although not essential for the analysis, this alternative color scheme would make the geographical patterns (or lack there of) to be more evident in the DiWANN networks.
7) In the Discussion section, the authors state that "MSA results indicate that the virus has been mutating faster than before". Is there evidence that these new variants are more prone to mutations than the initial variants? In other words, is there a greater rate of mutations in later time periods as compared to initial time periods? Or does the increased number of mutations observed in later time periods simply reflects the effect of the same mutation rate as before, but operating in a much larger set of infected individuals.
8) The authors single out 3 mutations that have occurred persistently in large number of sequences, namely L452R, E484K and N501Y. Is there anything else that this study can discuss about these mutations? Are they associated with any temporal or geographical patterns? Have these mutations been associated with any particular phenotypic trait or disease outcome?
9) The authors discussed the issue of how a poorly chosen threshold can affect the network analysis. Is there any insights from this study that can be generalized to similar analyses for other proteins/viruses? Or the threshold in each application case must be defined by trial and error, and a subjective sense of "balance" in the density of the network? Are there metrics that can be used to help determine if a chosen threshold is reasonable or detrimental for the analysis?
